Understanding the spread of de novo and transmitted macrolide-resistance in Mycoplasma genitalium

Cadosch Dominique 1
Garcia Victor 1 2
Jensen Jørgen S. 3
Low Nicola 1
Althaus Christian L. christian.althaus@alumni.ethz.ch 1
1 Institute of Social and Preventive Medicine, University of Bern , Bern , Switzerland
2 School of Life Sciences and Facility Management, Zurich University of Applied Sciences , Wädenswil , Switzerland
3 Research Unit for Reproductive Tract Microbiology, Statens Serum Institut , Copenhagen , Denmark
Nishiura Hiroshi
Electronic publication date: 2020 Apr 7
Publication date: 2020
Volume: 8
Electronic Location ID: e8913
Received 2019 Oct 29; Accepted 2020 Mar 15
Copyright: ©2020 Cadosch et al.
Copyright year: 2020
Copyright holder: Cadosch et al.
License: This is an open access article distributed under the terms of the Creative Commons Attribution License, which permits unrestricted use, distribution, reproduction and adaptation in any medium and for any purpose provided that it is properly attributed. For attribution, the original author(s), title, publication source (PeerJ) and either DOI or URL of the article must be cited.
License URL: https://creativecommons.org/licenses/by/4.0/

Keywords: Mycoplasma genitalium, Mathematical model, Antibiotic resistance, Sexually transmitted infection

Funding: Swiss National Science Foundation through the Epidemiology and Mathematical Modelling in Infectious Diseases Control (EpideMMIC) project 32003B 160320 This work was supported by the Swiss National Science Foundation through the Epidemiology and Mathematical Modelling in Infectious Diseases Control (EpideMMIC) project (grant number 32003B 160320). The funders had no role in study design, data collection and analysis, decision to publish, or preparation of the manuscript.

==============================
Background

The rapid spread of azithromycin resistance in sexually transmitted Mycoplasma genitalium infections is a growing concern. It is not yet clear to what degree macrolide resistance in M. genitalium results from the emergence of de novo mutations or the transmission of resistant strains.

Methods

We developed a compartmental transmission model to investigate the contribution of de novo macrolide resistance mutations to the spread of antimicrobial-resistant M. genitalium. We fitted the model to resistance data from France, Denmark and Sweden, estimated the time point of azithromycin introduction and the rates at which infected individuals receive treatment, and projected the future spread of resistance.

Results

The high probability of de novo resistance in M. genitalium accelerates the early spread of antimicrobial resistance. The relative contribution of de novo resistance subsequently decreases, and the spread of resistant infections in France, Denmark and Sweden is now mainly driven by transmitted resistance. If treatment with single-dose azithromycin continues at current rates, macrolide-resistant M. genitalium infections will reach 25% (95% confidence interval, CI [9–30]%) in France, 84% (95% CI [36–98]%) in Denmark and 62% (95% CI [48–76]%) in Sweden by 2025.

Conclusions

Blind treatment of urethritis with single-dose azithromycin continues to select for the spread of macrolide resistant M. genitalium. Clinical management strategies for M. genitalium should limit the unnecessary use of macrolides.

Introduction

Macrolide-resistant Mycoplasma genitalium poses a considerable problem for clinical practice and public health, with more than 40% of detected infections being resistant in several countries (Gesink et al., 2012; Pond et al., 2014; Salado-Rasmussen & Jensen, 2014; Murray et al., 2017). M. genitalium is a sexually transmitted bacterium, which is often asymptomatic and, untreated, persists for more than a year (Smieszek & White, 2016; Cina et al., 2019). The prevalence of M. genitalium in the general population aged 16 to 44 years has been estimated to be 1.3% (95% confidence interval, CI [1.0–1.8]%) and 3.9% (95% CI [2.2–6.7]%) in countries with higher and lower levels of development (Baumann et al., 2018), and is similar in women and men. Like Chlamydia trachomatis, M. genitalium causes non-gonococcal urethritis (NGU) in men (Taylor-Robinson & Jensen, 2011) and lower and upper genital tract disease in women (Wiesenfeld & Manhart, 2017).

M. genitalium was first isolated in 1980 from two men with NGU (Tully et al., 1981), but it has fastidious growth requirements, is slow-growing and difficult to culture (Taylor-Robinson & Jensen, 2011), hampering the progress of clinical research. Reliable detection, first by polymerase chain reaction and subsequently other nucleic acid amplification tests (NAATs), was not possible until the early 1990s, (Gaydos, 2017). Most currently used diagnostic tests do not detect resistance mutations, but commercial assays that can provide information on macrolide resistance have become available (Unemo & Jensen, 2017). In most clinical settings, however, NAATs for M. genitalium diagnosis are still not available. The clinical syndrome of NGU is therefore often treated empirically, with a single 1g dose of azithromycin recommended for first line treatment in many countries since the late 1990s (Bradshaw, Jensen & Waites, 2017).

Macrolide resistance in M. genitalium results from a single nucleotide mutation in region V of the 23S rRNA gene, most commonly A2058G or A2059G. Jensen et al. (2008) identified these mutations in Australian and Swedish men, with NGU caused by M. genitalium, who experienced clinical treatment failure with 1g azithromycin. The men carried a wild-type organism before treatment, but post-treatment specimens contained mutations in the 23S rRNA gene that conferred macrolide resistance. Since then, other investigators have detected macrolide resistance mutations de novo (also known as acquired, induced or selected) in M. genitalium (Ito et al., 2011; Twin et al., 2012; Anagrius, Loré & Jensen, 2013; Bissessor et al., 2015; Couldwell et al., 2013; Walker et al., 2013; Falk, Enger & Jensen, 2015; Read et al., 2017), and a meta-analysis of studies published up to 2016 estimated a 12.0% (95% CI [7.1–16.9]%) probability of de novo resistance after treatment with 1g of azithromycin (Horner et al., 2018). Once acquired, untreated resistant strains can be transmitted to new sexual partners.

Recommendations for future research on M. genitalium prioritize the need for more effective and safe antimicrobials (Martin, Manhart & Workowski, 2017). It is important to understand the degree to which treatment failure in M. genitalium results from the emergence of de novo resistance mutations or the transmission of resistant strains because the type of resistance will influence future treatment strategies. The objective of this study was to investigate the role of de novo and transmitted resistance in the spread of azithromycin-resistant M. genitalium.

Methods

We developed a mathematical model of M. genitalium transmission and fitted it to epidemiological data about time trends in macrolide resistance. We define ‘de novo’ as a change from a drug-sensitive infection before treatment to a drug-resistant infection after treatment, either by selection of one or a few pre-existing resistant mutants in an otherwise drug-sensitive bacterial population or due to a novel resistance mutation evolving during drug exposure. Mathematical modeling and parameter inference were conduced in the R software environment for statistical computing (R Core Team, 2016). All code files for the transmission model are available on GitHub (https://github.com/calthaus/MG-resistance).

Epidemiological data

We searched Pubmed up to May 4, 2018. We used the medical subject headings Mycoplasma genitalium AND drug resistance, bacterial and found 67 publications. From these, two authors independently selected six studies for three countries that met the following criteria: country with multiple studies that reported on M. genitalium and macrolide resistance mutations, data for more than three years from the same region or the entire country, and use of different strategies to test and treat M. genitalium. For each country, we recorded the testing strategy and treatment regimen, year in which azithromycin was introduced for M. genitalium treatment, numbers of specimens with positive results for M. genitalium and the number with macrolide resistance mutations. We contacted study authors for additional information. For each year, we calculated the proportion (with 95% CI) of azithromycin-resistant M. genitalium.

Transmission model

We developed a deterministic, population-based compartmental model that describes the spread of drug resistant M. genitalium (Fig. 1, Table 1). The model consists of four compartments: susceptibles (S), people infected with a drug-sensitive strain of M. genitalium (IS), and people infected with a drug-resistant strain of M. genitalium that was either acquired during treatment (IA) or transmitted (IT). Assuming a homogenous population without demography, the transmission dynamics can be described by the following set of ordinary differential equations (ODEs): (1) dSdt=−βSIS+IA+IT+γIS+IA+IT+1−μτIS,

(2) dISdt=βSIS−γIS−τIS,

(3) dIAdt=μτIS−γIA,

(4) dITdt=βSIA+IT−γIT,

where β is the transmission rate, which is assumed to be the same for both strains of M. genitalium. Both types of infections can clear naturally at rate γ. Patients receive treatment at rate τ. The treatment rate is defined as all occasions of treatment with a single 1g dose of azithromycin in a person infected with M. genitalium, either with or without symptoms. µdenotes the probability of de novo resistance emergence during treatment. The de novo emergence of resistance also implies that the treatment failed. We used the point estimate of the probability of de novo resistance emergence of 12% from Horner et al. (2018). For simplicity, we assumed that resistant infections only clear naturally, with no second-line treatment.

Figure 1 Structure of the transmission model for Mycoplasma genitalium.

Table 1 Parameters of the transmission model for Mycoplasma genitalium.

CI: confidence intervals.

Parameter	Description	Value (95% CI)	Reference or comment	
β	Transmission rate	0.816 person−1 y−1	Calibrated to prevalence	
γ	Natural clearance rate	0.8 y−1	Smieszek & White (2016)	
τ	Treatment rate of	0.04 y−1 (0.03–0.04 y−1)	Model estimate: France	
	infected individuals	0.13 y−1 (0.05–0.34 y−1)	Model estimate: Denmark	
		0.14 y−1 (0.11–0.18 y−1)	Model estimate: Sweden	
µ	Probability of de novo	12%	Horner et al. (2018)	
	resistance during treatment			

In the transmission model, drug-sensitive (IS) and drug-resistant (IA and IT) M. genitalium strains compete for the same resource, i.e., the susceptible hosts (S). The rate at which the resistant strain replaces the sensitive strain can be expressed by the difference in their net growth rates (Δϕ) (Bonhoeffer, Lipsitch & Levin, 1997; Fingerhuth et al., 2016): (5) Δϕ=ϕA+T−ϕS=dIAdt+dITdtIA+IT−dISdtIS=βS−γ+μτISIA+IT−βS−γ−τ=τ1+μISIA+IT=τ1+μ1−pp,

where p denotes the proportion of resistant infections among all infections. Note that Δϕ does not depend on the transmission rate β or the natural clearance rate γ, i.e., is unaffected by the overall prevalence of M. genitalium.

Model parameters

We set the natural clearance rate (γ) of M. genitalium to 0.8 y−1 (Smieszek & White, 2016). We calibrated the transmission rate β to 0.816 person−1 y−1, which results in an equilibrium prevalence of 2% in the absence of treatment and is consistent with estimates of the prevalence of M. genitalium in sexually active adults in high-income countries (Baumann et al., 2018). The values for the transmission rate and the natural clearance rate, and correspondingly the initial prevalence, do not govern the relative growth rate of the drug-resistant proportion (Δϕ), so they do not influence the relative prevalence of resistant infections or the model fits and parameter estimates. We did not find any published evidence of the effect of macrolide resistance on the fitness of M. genitalium strains, so we assumed that any fitness reduction is negligible and that resistant and wild-type strains have the same infectivity. The probability of emergence of de novo resistance during treatment (µ) was set to 12%, as reported in the meta-analysis by Horner et al. (2018).

Model fitting and simulations

We fitted the transmission model to country-specific resistance data to obtain maximum likelihood estimates of the treatment rate of infected people, τ, and the time point T for the introduction of azithromycin. Given a model-predicted proportion of resistant strains pi=IAi+ITiISi+IAi+ITi in year i (Table 2), the binomial log-likelihood to find ki resistant samples in Ni tested individuals is (6) Lτ,T= ∑logNiki+ki logpi+Ni−kilog1−pi.

Table 2 Characteristics of studies with time trend data about azithromycin-resistant M. genitalium infections.

rRNA, ribosomal ribonucleic acid; MG, M. genitalium; RT-PCR, real-time PCR.

Reference	Study year	Setting	Study	Method of	Number of	Number of samples	Comments	
	or period		population	detection	MG positive	with mutations in		
					samples tested	23S rRNA gene		
Chrisment et al. (2012)	2003	Pellegrin Hospital,	Retrospective analysis of	RT-PCR and	1	0	Only 4 specimens	
	2004	Bordeaux, France;	MG-positive specimens from	sequencing	10	0	from Paris clinic	
	2005	Saint-Louis Hospital,	sexually transmitted disease		6	0		
	2006	Paris France	clinics and general practice		10	1		
	2007		clinics		15	2		
	2008				13	2		
	2009				21	3		
	2010				39	5		
Touati et al. (2014)	2011	Pellegrin Hospital,	Retrospective analysis of	RT-PCR and	69	10		
	2012	Bordeaux, France;	MG-positive specimens	high-resolution	65	9		
				melt analysis				
Le Roy et al. (2016)	2013	Bordeaux University	Retrospective analysis of	RT-PCR and	112	19		
	2014	Hospital, Bordeaux,	MG-positive specimens	high-resolution	109	19		
		France		melt analysis				
Le Roy et al. (2017)	2016	Bordeaux University	Prospective collected	RT-PCR and	72	6		
		Hospital, Bordeaux,	specimens from patients	high-resolution				
		France		melt analysis				
Salado-Rasmussen & Jensen (2014)	2007	General practitioners,	Retrospective analysis of	RT-PCR and	11	3	Data for individual years	
	2008	private specialists, and	MG-positive specimens	rapid	226	81	were aggregated in the	
	2009	hospitals across Denmark		pyrosequencing	378	135	publication. Statens Serum	
	2010				454	191	Institut was only laboratory	
							testing for macrolide resistance.	
Anagrius, Loré & Jensen (2013)	2006	Department of Venerology,	Retrospective analysis of	RT-PCR and	18	0	Study authors provided patient	
	2007	Central Hospital, Falun,	MG-positive specimens	sequencing	53	0	numbers for each year and data	
	2008	Sweden			58	1	for 2012 and 2013.	
	2009				81	5		
	2010				98	14		
	2011				100	21		
	2012				71	8		
	2013				114	10		

Simulations start at time T with 98% uninfected people, 2% people with drug-susceptible infections and no drug-resistant infections. We used log-transformed parameters for the estimation and stipulated that the lower and upper limits of T could not be before 1990 or after the time point when resistance was first observed. We derived simulation-based 95% CIs for the model curve from 10,000 bootstrap samples from the multivariate normal distribution of the two parameters using the R package mvtnorm. We used the ode function from the package deSolve to solve the ODEs, and the mle2 function from the package bbmle using the Nelder–Mead method for log-likelihood optimization.

To investigate the influence of the level of de novo resistance emergence on the rapid rise in the proportion of resistant infections, we simulated two alternative scenarios. In these scenarios, we kept the model-derived maximum likelihood estimates of τ and T but set the probability of de novo resistance emergence to lower values (µ= 1% and µ= 0.1%).

Results

Description of the data

We included six studies that provided data about the proportion of azithromycin-resistant M. genitalium infections over time and the management of M. genitalium infection in France (Chrisment et al., 2012; Touati et al., 2014; Le Roy et al., 2016; Le Roy et al., 2017), Denmark (Salado-Rasmussen & Jensen, 2014), and Sweden (Anagrius, Loré & Jensen, 2013) (Table 2). Study authors provided additional information from Denmark (data disaggregated by year) and Sweden (numbers of patients per year and unpublished data for 2012 and 2013).

In France, we included four studies with data from 542 samples from 2003 to 2016 (Chrisment et al., 2012; Touati et al., 2014; Le Roy et al., 2016; Le Roy et al., 2017). None of 17 M. genitalium positive specimens from 2003 to 2005 contained macrolide resistance mutations. From 2006 onwards, mutations were detected in 8% to 17% of specimens tested in each year. In France, azithromycin was introduced for first line treatment of NGU in the 1990s (Joly-Guillou & Lasry, 1999). For Denmark, one study reported nationwide data from 1,008 patients with M. genitalium detected from 2006 to 2010, with 27% to 42% of specimens containing macrolide resistance mutations (Salado-Rasmussen & Jensen, 2014). In Denmark, 1g single dose azithromycin is routinely prescribed for treatment of NGU; erythromycin was the first-line treatment before azithromycin became available. An extended azithromycin regimen is prescribed if a M. genitalium infection was diagnosed and NAAT for detection of M. genitalium infections have been available since 2003 (Salado-Rasmussen & Jensen, 2014). In Sweden, we analyzed one study with data about macrolide resistance mutations from 408 samples obtained from 2006 to 2013 from patients at a single clinic in Falun (Anagrius, Loré & Jensen, 2013). Macrolide resistance mutations were first detected in a single specimen in 2008 and increased to 16% of 95 specimens in 2011. In Sweden, doxycycline is used as first line treatment for NGU (Björnelius, Magnusson & Jensen, 2017). Azithromycin is used only when M. genitalium is identified as the cause, with testing introduced in the 2000s (Anagrius, Loré & Jensen, 2013).

Mathematical modeling

The transmission model fitted the increase in M. genitalium resistance in France, Denmark and Sweden well (Figs. 2A–2C). The model estimated treatment rates of infected people and dates of introduction of azithromycin were: France, treatment rate of 0.04 y−1 (95% CI [0.03–0.04] y−1), introduction of azithromycin in 1990 (95% CI [1990–2006]); Denmark, treatment rate of 0.13 y−1 (95% CI [0.05–0.34] y−1), introduction of azithromycin in 1995 (95% CI [1990–2006]); Sweden, treatment rate of 0.14 y−1 (95% CI [0.11–0.18] y−1), introduction of azithromycin in 2006 (95% CI [2005–2007]). A treatment rate of 0.14 y−1, such as in Sweden, corresponds to a proportion of 1 − e−0.14 ≈ 13% of infected individuals that will have received treatment after one year. If treatment with single-dose azithromycin continues at the estimated rates, macrolide-resistant M. genitalium infections will reach 25% (95% CI [9–30]%) in France, 84% (95% CI [36–98]%) in Denmark and 62% (95% CI [48–76]%) in Sweden by 2025.

Figure 2 Maximum-likelihood fits of the M. genitalium transmission model to data of azithromycin resistance in France, Denmark and Sweden.

)A–C) Increase in the proportion of drug-resistant M. genitalium infections. (D–F) Proportion of de novo resistance among all drug-resistant M. genitalium infections. Error bars and shaded areas correspond to the 95% confidence intervals of the data and model, respectively.

The importance of de novo resistance emergence for the early spread of macrolide-resistant M. genitalium becomes apparent in the alternative scenarios. Lower probabilities of de novo resistance, at the same estimated treatment rates and time points for the introduction of azithromycin as in the main model, would have resulted in considerably lower proportions of resistant infections (Figs. 2A, 2B and 2C). The influence of de novo resistance emergence on the rate of resistance spread can be explained by Eq. (5) (Fig. 3). As long as the proportion of resistant infections (p) is low, the contribution of de novo resistance emergence (µ) to the rate at which the resistant strain replaces the susceptible strain (Δϕ) is high. With increasing levels of the resistant strain, its growth advantage diminishes and slowly approaches Δϕ = τ, i.e., the spread of resistant infections will mainly be driven by transmitted resistance. This transition is depicted in Figs. 2D, 2E and 2F. At the time of introduction of azithromycin, the proportion of de novo resistance started at 100% and subsequently dropped in France, Denmark and Sweden. Since around 2015, the proportion of de novo resistance among all circulating macrolide-resistant M. genitalium infections has been low in all three countries.

Figure 3 Relative growth rate of drug-resistant M. genitalium infections as a function of the proportion of resistant infections. Lines show growth rates for the best fit models for France, Denmark and Sweden, assuming a probability of de novo resistance during treatment of µ= 12%.

Black horizontal lines correspond to the estimated treatment rates (τ) in each country.

Discussion

In this study, we fitted a compartmental transmission model to time trend data about the proportions of azithromycin-resistant M. genitalium infections in France, Denmark and Sweden, estimated the treatment rates and the time point of introduction of azithromycin, and projected that a majority of infections could become resistant to azithromycin in Denmark and Sweden by 2025. We further showed that de novo resistance emergence accelerated the early spread of macrolide-resistant M. genitalium, whereas the spread of resistant infections is now mainly driven by transmitted resistance.

A major strength of this study is the combination of empirical data sources and mathematical modeling. Parameters that were not available in the literature were indirectly inferred by fitting the model to observational data. Despite its simplicity, the model assumptions provide a coherent qualitative and quantitative explanation for the clinically observed rapid rise of macrolide-resistant M. genitalium infections.

There are some caveats to both the observational data sources and the model. First, owing to the small number of samples for each data point, particularly for early years, confidence intervals for the estimates of the proportion of resistant infections are wide. In Denmark, azithromycin has been used for a long time but data about the prevalence of drug resistant infections were only available since 2006, which introduces more uncertainty in the estimated point at which resistance emerged. Second, the characteristics of people tested for M. genitalium in the three countries are not well described and differences in testing practices between countries might account for some of the variation in the proportions with macrolide resistance. An increase over time in the proportion of resistant infections was, however, observed in all three countries. We made a number of simplifying assumptions in our transmission model. First, we assumed that treatment rates of infected individuals in each country were constant over time. Even though the use of azithromycin might have changed over time, a sensitivity analysis showed that a model with a stepwise change in the treatment rate does not improve the model fits (results not shown). Second, we assumed that no second-line treatments were used for resistant M. genitalium infections. In practice, since most M. genitalium infections are asymptomatic and diagnostic testing is still uncommon, we do not think that this simplification affected our results. Third, our model does not include detailed population structure because the rate at which drug-resistant bacterial strains spread in a population relative to drug-sensitive strains can often be explained by the treatment rate, rather than the sexual network structure (Fingerhuth et al., 2016). More complex models with different sexes, partner change rates and age structure, would be necessary to obtain a better description of the absolute prevalence of infections and resistance, but this was not the objective of this study.

Our study strongly suggests that, rather than resulting in ‘occasional treatment failure’ as originally believed (Jensen et al., 2008), the development of de novo resistant mutations in about one in eight M. genitalium infections (Horner et al., 2018) is a major driver of azithromycin resistance during the early phase of resistance spread. This finding is supported by data from France and Sweden (Anagrius, Loré & Jensen, 2013; Chrisment et al., 2012; Touati et al., 2014; Le Roy et al., 2016; Le Roy et al., 2017), where no macrolide resistant mutations were detected initially, but a substantial proportion of diagnosed M. genitalium infections were azithromycin-resistant after just a few years of azithromycin use. The contribution of de novo resistance emergence to the spread of resistant infections decreases as the proportion of resistant infections increases. Our model-predicted estimates of the introduction of azithromycin for the treatment of NGU were consistent with published data describing its use in France (Joly-Guillou & Lasry, 1999) and Denmark in the 1990s, but later introduction in Sweden (Anagrius, Loré & Jensen, 2013). Our estimated treatment rate of infected individuals for France was lower than those for Denmark and Sweden. The estimated rates in Denmark and Sweden are comparable to those estimated in another epidemiological model of M. genitalium infections in the United Kingdom (Birger et al., 2017).

The high probability of de novo emergence of macrolide resistance mutations during treatment of M. genitalium infections appears to differ from experiences with some other sexually transmitted bacterial infections. A 1g dose of azithromycin might often be insufficient to eradicate a M. genitalium infection in concert with host immune responses, allowing for either a resistance mutation to occur in the single 23S rRNA operon during treatment or the survival of a few pre-existing drug-resistant bacteria and the subsequent selection of the mutants. The latter explanation is favored by the strong association with de novo resistance and high organism load (Bissessor et al., 2015; Read et al., 2017), but both mechanisms may play a role. The high probability of de novo resistance also has implications for antimicrobial stewardship, as reducing blind treatment of urethritis with single dose azithromycin could potentially recover drug susceptibility. However, this would only be expected in the presence of a fitness cost, which has not been observed for macrolide resistance in M. genitalium. The absence of an observable fitness cost, or of routine tests to detect macrolide resistance mutations, has resulted in the rapid emergence and spread of M. genitalium resistance. In contrast, selection pressure exerted by treatment and clonal spread are the major drivers of the spread of macrolide-resistant Neisseria gonorrhoeae, with de novo resistance considered to be negligible (Fingerhuth et al., 2016). N. gonorrhoeae has four copies of the 23S rRNA gene and resistance increases with the number of mutated copies (Unemo & Shafer, 2014). In addition, active measures are used to limit the potential for the emergence of de novo macrolide resistance in N. gonorrhoeae, including dual therapy, in which azithromycin is a second drug in combination with ceftriaxone. Transmitted resistance is assumed to be responsible for most antimicrobial resistance, but a high rate of de novo resistance emergence has been observed during treatment with various antibiotics of infections such as Pseudomonas aeruginosa and Enterobacteriaceae (Chow et al., 1991; Carmeli et al., 1999). In general, de novo selection of drug-resistant mutants within a single patient occurs more often if the resistance is mediated by single-base mutations than if acquisition of efflux pumps or other complex mechanism are needed (Unemo & Jensen, 2017). Thus, de novo resistance is distinct from the selection of drug resistance as a result of treatment at the population level, which is more often transmitted; a situation which is seen with most other bacterial and parasitic sexually transmitted infections.

Conclusions

Current management strategies for M. genitalium will result in a majority of infections becoming resistant to azithromycin within the next few years, posing considerable problems for clinical management and population level control strategies (Golden, Workowski & Bolan, 2017). Screening and treatment of asymptomatic M. genitalium with 1g azithromycin regimens will further drive the spread of either de novo or transmitted resistance in countries with low or high levels of resistance, with absent evidence of a reduction in clinical morbidity (Golden, Workowski & Bolan, 2017). Treatment strategies to maintain the use of existing antimicrobials are now being evaluated since resistance to second line treatment with moxifloxacin is already increasing (Murray et al., 2017). In an observational study, resistance-guided therapy for symptomatic M. genitalium, with initial treatment with doxycycline followed by 2.5 g azithromycin over three days for macrolide susceptible infections and sitafloxacin for resistant infections resulted in an incidence of de novo macrolide resistance of 2.6% (95% CI [0.3–9.2]%) (Read et al., 2019). Randomized controlled trials are now needed to evaluate different treatment algorithms and new antimicrobials or combination therapy that might have a lower propensity for the emergence of de novo resistance (Bradshaw, Jensen & Waites, 2017). Blind treatment of urethritis with single dose azithromycin, which induces de novo resistance and selects for transmitted resistance in M. genitalium, is not recommended. Clinical management strategies for M. genitalium and other STIs should seek to limit the unnecessary use of macrolides.

We would like to thank Carin Anagrius from the Falu lasarett in Falun, Sweden and Kirsten Salado-Rasmussen from the Bispebjerg Hospital in Copenhagen, Denmark for providing us with additional unpublished data.

Additional Information and Declarations

Competing Interests

Author Contributions

Data Availability

Christian L. Althaus is an Academic Editor for PeerJ.

Dominique Cadosch conceived and designed the experiments, performed the experiments, analyzed the data, prepared figures and/or tables, authored or reviewed drafts of the paper, and approved the final draft.

Victor Garcia and Jørgen S. Jensen analyzed the data, authored or reviewed drafts of the paper, and approved the final draft.

Nicola Low conceived and designed the experiments, analyzed the data, authored or reviewed drafts of the paper, and approved the final draft.

Christian L. Althaus conceived and designed the experiments, performed the experiments, analyzed the data, prepared figures and/or tables, authored or reviewed drafts of the paper, and approved the final draft.

The following information was supplied regarding data availability:

All code files for the transmission model are available on GitHub: https://github.com/calthaus/MG-resistance.

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
