# Peer review of "Understanding the spread of de novo and transmitted macrolide-resistance in Mycoplasma genitalium"

_PeerJ, doi:10.7717/peerj.8913_

## Round 0.1 · original submission · Major Revisions

Two independent reviewers plus the handling editor (myself) have assessed your manuscript. The validity of model is ok, and so, please pay attention to stylistic comments from the reviewer to polish your manuscript.

My own review comments are below. Please note that I suggested a small piece of sensitivity analysis as an additional effort.


Basic reporting

Well written paper. The scope and objectives are clearly written.

Experimental design

[important remark] The time of introduction $T$ is sought by the model, but the initial prevalence is fixed. The simulation is starting with the boundary of 2% prevalence, meaning that $R_0$ is fixed automatically. Don't we have to examine the impact of different transmissibility on the resulting proportion of de novo transmission?

[important remark 2] The treatment rate $\tau$ is statistically estimated by country. While it's interesting to see different values of $\tau$ by country, the model fit is not perfect, and it is strange to argue that azithromycin treatment rate has been kept constant over time. Wasn't there any change point of azithromycin use in those countries? For instance, when was it introduced to the treatment guideline? When was it covered by treatment formulae and covered by insurance? At least, based on these pieces of information, one can try to use a step function to replace the constant $\tau$, as part of sensitivity analysis.

Validity of the findings

Validity is ok. I can reproduce similar results.

Comments for the author

Page 1. Please discuss clinical signs and symptoms in Intro. Also, please describe the burden of disease (can be incidence/mortality and can be cost).

Page 2, Intro: Please explicitly describe that the large proportion being de novo emergence has a very important implication to AMS. Namely, tapering the use of azithromycin can expect the recovery of drug susceptibility.

Page 5, equation (6): Please indicate in what range $i$ was summed.

Reviewer 1 ·

Basic reporting

This paper is well written in clear English with appropriate referencing.

In the introduction, I suggest that you add a little bit more information about Mycoplasma genitalium infection, including epidemiology and natural history.

Line 157: remove "model"

Line 158 : (Figure 3, left panels), Fig 2?

Figure 3: Horizontal lines look like black color.

Experimental design

Your compartment model is simple and easy to understand, although this model is oversimplified.

I disagree with the equation 5. φA+T is not the growth rate of the resistant strain because the rate itself includes the prevalence of the resistant strain.

In the equation 6, what is the meaning of "find ki resistant samples in Ni tested individuals"?

Validity of the findings

Table 1: I suggest you to change the column name " number of specimens tested" . because it was a bit confusing. (e.g. number of mycoplasma genitalium positive)

Please explain a little bit more about fig 2 right panels

·

Basic reporting

1. Your first sentence in introduction(lines 32-34) needs more detail. For example, '~ a short history of macrolide usage as treatment of the organism'.
2. The latter part of same paragraph (lines 37-40) need more explanation. It seems NAATs are being used but also not being used at same time to detect M. geitalium.

Experimental design

1. Please clarify which criteria did you refer to while you include or exclude the data.(lines 71-76)

Validity of the findings

1. In the graph of France in the right part Figure 2, the line is barely within the shade. Please provide more data from France which support your results.

Additional comments

Your results are very impressive. Your model is simple and well explaining.
More comprehensive introduction and methods would make this article greater.

---

## Round 0.2 · accepted · Accept

Authors should be congratulated on appropriate revisions.

Reviewer 1 ·

Basic reporting

Well written

Experimental design

Good

Validity of the findings

Good

Additional comments

After reviewing the updated version of the manuscript, I think this manuscript is ready for publication.

---

## Author Rebuttal · Round 0.2

Response to the reviewers' comments on the manuscript *"Understanding the spread of de novo and transmitted macrolide-resistance in Mycoplasma genitalium"* (#2019:10:42504:0:0:REVIEW) submitted to PeerJ.

We would like to thank the editor and the reviewers for carefully assessing our manuscript. Their constructive comments helped us to improve our manuscript. We hope that we have satisfactorily dealt with the comments, and that our revised manuscript is now acceptable for publication.

The page and line numbers refer to the revised version of our manuscript. We also provide a separate manuscript file which has all changes underlined and highlighted in blue.

**Editor comments (Hiroshi Nishiura):**

> *"Basic reporting*
>
> *Well written paper. The scope and objectives are clearly written.*
>
> *Experimental design*
>
> *[important remark] The time of introduction $T$ is sought by the model, but the initial prevalence is fixed. The simulation is starting with the boundary of 2% prevalence, meaning that $R_0$ is fixed automatically. Don't we have to examine the impact of different transmissibility on the resulting proportion of de novo transmission?"*

Different transmission rates ($\beta$) that result in different initial prevalence levels do not have an impact on the proportion of resistant infections (Fig. 2, left panels) nor the proportion of *de novo* resistance (Fig. 2, right panels). While this result is perhaps somewhat counterintuitive at first, Eq. 5 illustrates that the rate at which the resistant strain replaces the sensitive strain is independent of parameters that determine the initial prevalence or $R_0$. Thus, assuming a different transmission rate (and correspondingly a different initial prevalence) results in exactly the same model fit as in Fig. 2. We added the following sentence to further highlight this dynamic property (p. 4, lines 101-103): "Note that $\Delta\varphi$ does not depend on the transmission rate $\beta$ or the natural clearance rate $\gamma$, i.e., is unaffected by the overall prevalence of *M. genitalium*." We further clarified the following sentence (p. 4, lines 108-112): "The values for the transmission rate and the natural clearance rate, and correspondingly the initial prevalence, do not govern the relative growth rate of the drug-resistant proportion ($\Delta\varphi$), so they do not influence the relative prevalence of resistant infections or the model fits and parameter estimates."

> *"[important remark 2] The treatment rate $\tau$ is statistically estimated by country. While it's interesting to see different values of $\tau$ by country, the model fit is not perfect, and it is strange to argue that azithromycin treatment rate has been kept constant over time. Wasn't there any change point of azithromycin use in those countries? For instance, when was it introduced to the treatment guideline? When was it covered by treatment formulae and covered by insurance? At least, based on these pieces of information, one can try to use a step function to replace the constant $\tau$, as part of sensitivity analysis."*

We agree that the assumption of a constant treatment rate is a limitation of our model. We do not have detailed data on the use of azithromycin in the countries. However, we conducted a sensitivity analysis to investigate whether a model with a stepwise change in the treatment rate provides a better description of the data. To this end, we extended the model such that the treatment rate can switch from $\tau_1$ to $\tau_2$ at any time point after the introduction of azithromycin.

We found that the extended model does not improve the model fits, i.e., does not reduce the negative log-likelihood, compared to the baseline model with a single treatment rate except under the following scenarios: $\tau_2 = 0.0$ y$_{-1}$ (no treatment) in France after 2009, and a slight increase in $\tau_2$ just before the last data point in Denmark. We found these scenarios unrealistic and therefore decided not to include the results of the sensitivity analysis in our revised manuscript. However, we extended our Discussion as follows (p. 9, lines 204-208): "First, we assumed that treatment rates of infected individuals in each country were constant over time. Even though the use of azithromycin might have changed over time, a sensitivity analysis showed that a model with a stepwise change in the treatment rate does not improve the model fits (results not shown)."

> *"Validity of the findings*
>
> *Validity is ok. I can reproduce similar results.*
>
> *Comments for the author*
>
> *Page 1. Please discuss clinical signs and symptoms in Intro. Also, please describe the burden of disease (can be incidence/mortality and can be cost)."*

We extended the first two paragraphs of the Introduction as follows (p. 1-2, lines 32-53): "Macrolide-resistant *Mycoplasma genitalium* poses a considerable problem for clinical practice and public health, with more than 40% of detected infections being resistant in several countries (Gesink et al., 2012; Pond et al., 2014; Salado-Rasmussen and Jensen, 2014; Murray et al., 2017). *M. genitalium* is a sexually transmitted bacterium, which is often asymptomatic and, untreated, persists for more than a year (Smieszek and White, 2016; Cina et al., 2019). The prevalence of *M. genitalium* in the general population aged 16 to 44 years has been estimated to be 1.3% (95% confidence interval, CI: 1.0-1.8%) and 3.9% (95% CI: 2.2-6.7%) in countries with higher and lower levels of development (Baumann et al., 2018), and is similar in women and men. Like *Chlamydia trachomatis*, *M. genitalium* causes non-gonococcal urethritis (NGU) in men (Taylor-Robinson and Jensen, 2011) and lower and upper genital tract disease in women (Wiesenfeld and Manhart, 2017).

*M. genitalium* was first isolated in 1980 from two men with NGU (Tully et al., 1981), but it has fastidious growth requirements, is slow-growing and difficult to culture (Taylor-Robinson and Jensen, 2011), hampering the progress of clinical research. Reliable detection, first by polymerase chain reaction and subsequently other nucleic acid amplification tests (NAATs), was not possible until the early 1990s (Gaydos, 2017). Most currently used diagnostic tests do not detect resistance mutations, but commercial assays that can provide information on macrolide resistance have become available (Unemo and Jensen, 2017a). In most clinical settings, however, NAATs for *M. genitalium* diagnosis are still not available. The clinical syndrome of NGU is therefore often treated empirically, with a single 1g dose of azithromycin recommended for first line treatment in many countries since the late 1990s (Bradshaw et al., 2017)."

> *"Page 2, Intro: Please explicitly describe that the large proportion being de novo emergence has a very important implication to AMS. Namely, tapering the use of azithromycin can expect the recovery of drug susceptibility."*

We agree that this is an interesting implication, and decided to highlight this aspect in the Discussion (p. 10, lines 239-243): "The high probability of *de novo* resistance also has implications for antimicrobial stewardship, as reducing blind treatment of urethritis with single dose azithromycin could potentially recover drug susceptibility. However, this would only be expected in the presence of a fitness cost, which has not been observed for macrolide resistance in *M. genitalium*."

*"Page 5, equation (6): Please indicate in what range $i$ was summed."*

$i$ corresponds to the study year, as listed in Table 2. We added a reference to the table in the corresponding sentence.

**Reviewer 1 (Anonymous):**

*"Basic reporting*

*This paper is well written in clear English with appropriate referencing.*

*In the introduction, I suggest that you add a little bit more information about Mycoplasma genitalium infection, including epidemiology and natural history."*

We changed the Introduction accordingly (see our response to the Editor's similar comment above).

*"Line 157: remove "model""*

Corrected.

*"Line 158 : (Figure 3, left panels), Fig 2?"*

Corrected.

*"Figure 3: Horizontal lines look like black color."*

Corrected.

*"Experimental design*

*Your compartment model is simple and easy to understand, although this model is oversimplified.*

*I disagree with the equation 5. φA+T is not the growth rate of the resistant strain because the rate itself includes the prevalence of the resistant strain."*

Eq. 5 is correct and derives from classical population dynamics (also see Bonhoeffer et al., 1997, and Fingerhuth et al., 2016). The growth rate of the resistant strain ($I_A + I_T$) does indeed depend on the ratio of the sensitive strain over the resistant strain ($I_S / (I_A + I_T)$) because of the probability of *de novo* resistance during treatment of the sensitive strain (also see Fig. 3). We extended Eq. 5 to indicate how the (per capita) net growth rates of the resistant and sensitive strain are derived.

*"In the equation 6, what is the meaning of "find ki resistant samples in Ni tested individuals"?"*

This refers to the binomial sampling process, which we now indicate in the definition of the likelihood.

*"Validity of the findings*

*Table 1: I suggest you to change the column name " number of specimens tested " . because it was a bit confusing. (e.g. number of mycoplasma genitalium positive)"*

Changed.

*"Please explain a little bit more about fig 2 right panels"*

We added the following text (p. 8, lines 177-181): "This transition is depicted in Fig. 2 (right panels). At the time of introduction of azithromycin, the proportion of *de novo* resistance started at 100% and subsequently dropped in France, Denmark and Sweden. Since around 2015, the proportion of *de novo* resistance among all circulating macrolide-resistant *M. genitalium* infections has been low in all three countries."

**Reviewer 2 (Ki-Deok Lee):**

*" Basic reporting*

*1. Your first sentence in introduction(lines 32-34) needs more detail. For example, '~ a short history of macrolide usage as treatment of the organism'."*

*2. The latter part of same paragraph (lines 37-40) need more explanation. It seems NAATs are being used but also not being used at same time to detect M. geitalium."*

We changed the Introduction accordingly (see our response to the Editor's similar comment above).

*" Experimental design*

*1. Please clarify which criteria did you refer to while you include or exclude the data.(lines 71-76)"*

We clarified the section as follows (p. 2-3, lines 83-87): "From these, two authors independently selected six studies for three countries that met the following criteria: country with multiple studies that reported on *M. genitalium* and macrolide resistance mutations, data for more than three years from the same region or the entire country, and use of different strategies to test and treat *M. genitalium*."

*"Validity of the findings*

*1. In the graph of France in the right part Figure 2, the line is barely within the shade. Please provide more data from France which support your results."*

Unfortunately, our review did not result in additional data for France. The uncertainty in the proportion of *de novo* resistance for France can be explained by the lower limit for the time point for the introduction of azithromycin *T*.

*"Comments for the Author*

*Your results are very impressive. Your model is simple and well explaining. More comprehensive introduction and methods would make this article greater."*

Thank you for your positive assessment of our manuscript.